# Usb1 controls U6 snRNP assembly through evolutionarily divergent cyclic phosphodiesterase activities

Allison L. Didychuk[1], Eric J. Montemayor[1,2], Tucker J. Carrocci[1], Andrew T. DeLaitsch[1], Stefani E. Lucarelli[1], William M. Westler[3], David A. Brow[2], Aaron A. Hoskins[1] & Samuel E. Butcher [1,3]

U6 small nuclear ribonucleoprotein (snRNP) biogenesis is essential for spliceosome assembly, but not well understood. Here, we report structures of the U6 RNA processing enzyme Usb1 from yeast and a substrate analog bound complex from humans. Unlike the human ortholog, we show that yeast Usb1 has cyclic phosphodiesterase activity that leaves a terminal 3′ phosphate which prevents overprocessing. Usb1 processing of U6 RNA dramatically alters its affinity for cognate RNA-binding proteins. We reconstitute the post-transcriptional assembly of yeast U6 snRNP in vitro, which occurs through a complex series of handoffs involving 10 proteins (Lhp1, Prp24, Usb1 and Lsm2–8) and anti-cooperative interactions between Prp24 and Lhp1. We propose a model for U6 snRNP assembly that explains how evolutionarily divergent and seemingly antagonistic proteins cooperate to protect and chaperone the nascent snRNA during its journey to the spliceosome.

---

[1] Department of Biochemistry, University of Wisconsin, Madison, Wisconsin 53706, USA. [2] Department of Biomolecular Chemistry, University of Wisconsin, Madison, Wisconsin 53706, USA. [3] National Magnetic Resonance Facility at Madison, Biochemistry Department, University of Wisconsin-Madison, Madison, Wisconsin 53706, USA. Correspondence and requests for materials should be addressed to S.E.B. (email: sebutcher@wisc.edu)

Splicing of precursor messenger RNA is an essential process in all eukaryotes and is catalyzed by the spliceosome. The spliceosome is a dynamic macromolecular machine composed primarily of five ribonucleoprotein particles known as the U1, U2, U4, U5 and U6 small nuclear ribonucleoproteins (snRNPs), each containing a small nuclear RNA (snRNA) and numerous proteins. The highly conserved U6 snRNA coordinates magnesium ions in the active site that are required for splicing catalysis[1]. Unlike the other snRNAs, U6 is synthesized by RNA polymerase III (Pol III) and, like other Pol III transcripts, its transcription is terminated stochastically when the polymerase encounters a stretch of adenines in the template strand (Fig. 1a)[2–4] with Saccharomyces cerevisiae requiring at least six sequential adenines in the template strand to terminate efficiently[5]. Nascent U6 terminates in a 3′ polyuridine tract of heterogeneous length (4–8 uridines) with terminal 2′ and 3′ hydroxyl groups (a cis-diol) (Fig. 1a) and is bound by the La protein (Lhp1 in S. cerevisiae)[6, 7]. However, the predominant form of mature U6 in vivo in most organisms characterized to date does not contain a terminal cis-diol and is not bound by La protein[8].

Post-transcriptional exonucleolytic processing of U6 is directed by U6 biogenesis protein 1 (Usb1)[9, 10], a 3′–5′ exonuclease belonging to the 2H phosphodiesterase superfamily of enzymes[9, 11]. Usb1 is an essential protein in S. cerevisiae[9, 12], but not in Schizosaccharomyces pombe[10] and metazoans[10, 11, 13]. In yeast, it is likely that immature U6 RNA is the only essential target of Usb1, as overexpression of this snRNA rescues deletion of yUsb1[11]. Human Usb1 is involved in processing precursors of

**Fig. 1** yUsb1 acts as a 3′–5′exonuclease and CPDase in vitro. **a** U6 snRNA is synthesized by RNA Polymerase III. Transcription termination produces a heterogeneous U6 with a 4–8 nucleotide U-tail. Processing by yUsb1 shortens the U-tail and leaves a phosphoryl group. **b** Usb1 removes nucleotides from the 3′ end of RNAs. The 5′-labeled U6 95–112+3U oligonucleotide cis-diol substrate (lane 2) is insensitive to CIP (lane 3) or T4 PNK (lane 4) treatment. Incubation with yUsb1 for 1 h results in a shorter product (lane 5). Similar reactivity of the product to both CIP (lane 6) and T4 PNK (lane 7) indicates that the product is a noncyclic phosphate. An alkaline hydrolysis ladder (lane 1) shows the mobility of oligonucleotide products of different lengths. (**c**, *top*) One-dimensional [31]P NMR spectra of 2′,3′-cUMP shows a single peak at 20 ppm. A 3′ UMP standard has a single peak at 3.4 ppm. When 2′,3′-cUMP is incubated with AtRNL, which leaves a 2′ phosphate[8], there is a single peak at 3.2 ppm. Incubation of 2′,3′-cUMP with yUsb1 produces a new signal at 3.4 ppm (**c**, *bottom*) Zoom of dashed region in top panel. **d** Time course of Usb1 processing on RNAs with different 3′ end modifications. yUsb1 is most active on RNA substrates with a cis-diol (lanes 1–4), less active on those with a 2′,3′-cyclic phosphate (>p; lanes 5–8) or 2′ phosphates (2′P; lanes 9–12), and is inactive on 3′ phosphate ends (3′P; lanes 13–16). **e** Model describing the dual activities of yUsb1

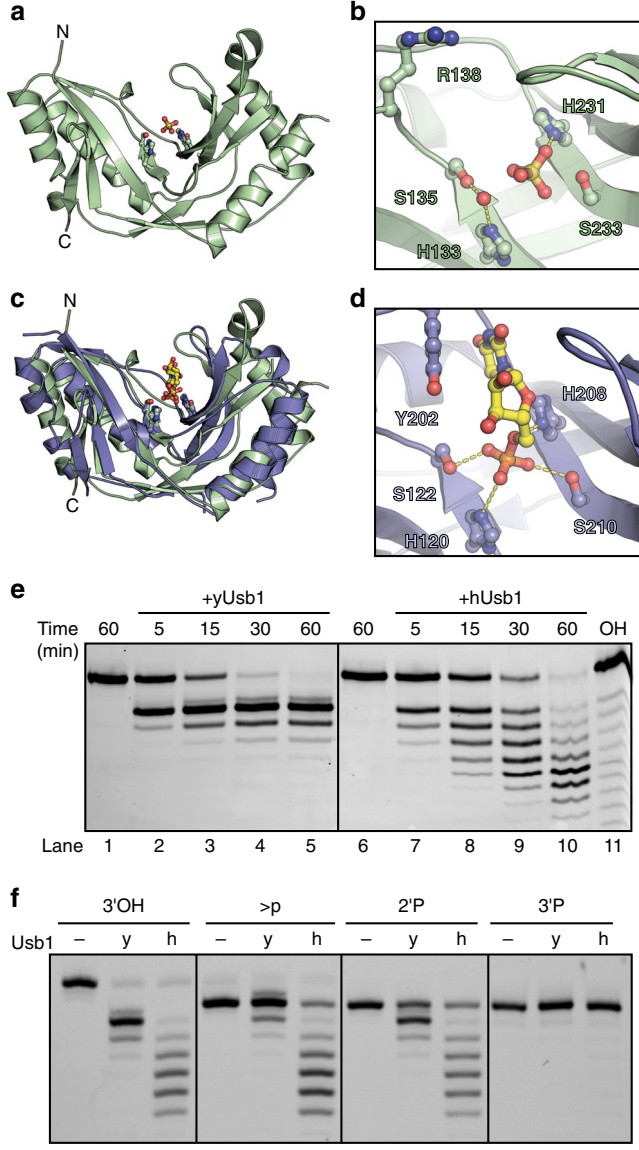

**Fig. 2** Structure of yUsb1 and structure of hUsb1 with a substrate analog bound in the active site. **a** Structure of yUsb1 at 1.8 Å. Residues 1–70 were excluded from the crystallizable construct as they were predicted to be unstructured. The two active site H-X-S motifs are shown in *ball-and-stick* representation. **b** Active site of yUsb1. A sulfate ion is coordinated in the center of the H-X-S motifs. A water molecule is coordinated by histidine 133 and serine 135. (**c**) Structure of hUsb1 with 5′ UMP in the active site (*blue*; 5′ UMP in *yellow*) is similar to the structure of yUsb1 (*green*). Residues 1–78 were truncated to facilitate crystallization. The two active site H-X-S motifs are shown in *sticks*. **d** Active site of hUsb1. 5′ UMP is coordinated in the center of the H-X-S motifs. The 5′ phosphate is positioned similarly to the sulfate ion in the yUsb1 structure (**b**). Tyrosine 202 makes a stacking interaction with the uracil of 5′ UMP. Hydrogen bonds between histidine 208 and the O5′ oxygen and between serine 122, histidine 120 and serine 210 and the phosphate hold 5′ UMP in the active site. **e** Time course comparing yUsb1 (lanes 1–5) and hUsb1 (lanes 6–10) activity. An alkaline hydrolysis ladder (lane 11) shows the mobility of oligonucleotide products of different length. **f** Single time-point (60 min) comparison of yUsb1 (y) and hUsb1 (h) on RNA substrates with different 3′ modifications. Both yUsb1 and hUsb1 are most active on 2′,3′-*cis*-diol RNAs (lanes 1–3), less active on cyclic phosphate (lanes 4–6) and 2′ phosphate (lanes 7–9) RNAs and inactive on 3′ phosphate RNAs

both U6 and the minor spliceosomes snRNA U6atac[13]. Loss-of-function mutations in human Usb1 are associated with poikiloderma with neutropenia, a rare skin disease that is also associated with loss of white blood cells[14].

In humans, processing of U6 by Usb1 creates a terminal 2′,3′-cyclic phosphate[11] which stimulates binding of Lsm2–8[15]. Binding of Lsm2–8 in turn promotes formation of the U6 and U4/U6 snRNPs[16–18]. Thus, the identity of the 3′ end of U6 RNA is a crucial determinant in the assembly of U6-containing snRNPs. However, the end modification of the terminal nucleotide in yeast differs from that in humans, with yeast primarily containing a noncyclic phosphate group[8]. Prior to this work, it was not known if yUsb1 directly promotes formation of the terminal phosphate, or if an additional cyclic phosphodiesterase (CPDase) acts on U6 RNA after exonucleolytic processing. Furthermore, the position of the terminal phosphate on yeast U6 RNA (at either the 2′ or 3′ oxygen) was not known, and cannot be resolved in recent cryo-EM structures due to local resolutions of >7 Å for the U6 3′ tail[19–24].

To investigate the mechanism of U6 RNA processing and snRNP assembly, we characterized the structure and activities of yeast and human Usb1. We determined the 1.8 Å crystal structure of the catalytic domain of yUsb1, and a 1.4 Å co-crystal structure of hUsb1 bound to the substrate analog uridine 5′-monophosphate (5′ UMP). We demonstrate the importance of the identity of the 3′ end of U6 to snRNP formation and show how U6 RNA is involved in a series of protein-mediated handoffs prior to formation of the mature U6 snRNP.

## Results

**Yeast Usb1 exhibits cyclic phosphodiesterase activity.** We sought to understand the activity of Usb1 from *S. cerevisiae* (yUsb1). To this end, we prepared full-length yUsb1 protein and tested it for exoribonuclease activity in vitro using oligonucleotide model substrates. When incubated with RNA terminating in multiple uridines and a *cis*-diol, yUsb1 predominately removed only 1 nucleotide from the 3′ end of an RNA, with 80% of the substrate converted into the *n-1* product (Fig. 1b). Interestingly, 15% of the *n-1* product had a slightly slower mobility consistent with it containing a cyclic phosphate 3′ end[25]. Removal of one or two additional nucleotides occurs infrequently (15 and 3% of the total product, respectively) (Fig. 1b). Substrates with a deoxyuridine at the *n-1* position were not processed by yUsb1 (Supplementary Fig. 1a), indicating that yUsb1 acts exclusively as a 3′–5′ exonuclease and that additional cleavage products are due to inefficient re-processing of *n-1* RNA. Additionally, we find that yUsb1 removes 1–3 nucleotides regardless of the length of the polyuridine tail (Supplementary Fig. 1b).

We investigated the product of yUsb1 processing by exploiting the disparate activities of calf intestinal phosphatase (CIP) and T4 polynucleotide kinase (PNK) (Fig. 1b). Both CIP and T4 PNK can remove terminal 3′ phosphoryl groups from oligonucleotides to produce *cis*-diols, but only T4 PNK can remove both cyclic and noncyclic terminal phosphates. Treatment of yUsb1-processed RNA with CIP or T4 PNK both resulted in reduced mobility of the products, consistent with the presence of a noncyclic phosphate group on yUsb1-processed RNA (Fig. 1b). Thus, yUsb1 directly catalyzes the formation of a noncyclic phosphate, and U6 3′ end processing does not require a *trans*-acting 2′,3′-cyclic phosphodiesterase in yeast. Yeast Usb1 catalyzes two distinct chemical reactions: (1) exonucleolytic removal of a terminal uridine ("first step") and (2) cyclic phosphodiesterase (CPDase) ring opening to leave a noncyclic phosphate ("second step").

To unambiguously identify whether yUsb1 leaves a 2′ or 3′ phosphate, we investigated the CPDase activity of yUsb1

**Table 1 Data collection and refinement statistics**

| | hUsb1 + 5′ UMP PDB 5V1M | yUsb1 PDB 5UQJ | yUsb1 phasing derivative |
|---|---|---|---|
| *Data collection* | | | |
| Wavelength (Å) | 0.9795 | 0.9786 | 0.9786 |
| Resolution range (Å)* | 53.2-1.47 (1.50-1.47) | 78.9-1.80 (1.84-1.80) | 81.9-2.00 (2.05-2.00) |
| Space group | $P2_1$ | $I4_122$ | $I4_122$ |
| Unit cell dimensions (Å) | 42.4, 53.2, 46.6 β=106.95° | 157.9, 157.9, 44.4 | 163.6, 163.6, 43.8 |
| Total reflections* | 224,991 (10,848) | 382,211 (21,365) | 597,214 (42,693) |
| Unique reflections* | 33,607 (1,621) | 26,187 (1,517) | 20,480 (1,464) |
| Multiplicity* | 6.7 (6.7) | 14.6 (14.1) | 29.2 (29.2) |
| Completeness (%)* | 99.3 (99.8) | 99.4 (98.0) | 100 (100) |
| Mean $I/\sigma(I)$* | 12.8 (1.2) | 23.3 (1.4) | 13.8 (1.1) |
| Wilson B-factor | 16.3 | 28.6 | 49.9 |
| *R*-merge* | 0.08 (1.51) | 0.09 (2.23) | 0.25 (4.84) |
| $CC_{1/2}$* | 0.999 (0.581) | 1.000 (0.446) | 0.999 (0.457) |
| | | | |
| *Refinement* | | | FOM=0.58 |
| $R_{work}/R_{free}$* | 0.16/0.19 (0.30/0.36) | 0.19/0.23 (0.34/0.39) | |
| Total number of atoms | 3,414 | 1,897 | |
| Macromolecules | 3,125 | 1,746 | |
| Ligands | 131 | 35 | |
| Water | 158 | 116 | |
| RMS (bonds) | 0.023 | 0.018 | |
| RMS (angles) | 2.062 | 1.681 | |
| Ramachandran favored | 98.4% | 95.7% | |
| Ramachandran outliers | 0% | 0.96% | |
| Average B factor (Å$^2$) | 28.1 | 36.1 | |
| Protein | 26.9 | 35.6 | |
| Ligands/ions | 42.9 | 46.7 | |
| Solvent | 38.7 | 41.2 | |

*Values shown in parentheses are for the highest resolution shell

in isolation from its exonuclease activity using nuclear magnetic resonance (NMR) spectroscopy (Fig. 1c). The $^{31}$P chemical shift for a 2′,3′-cyclic phosphate is ~ 20 ppm, whereas noncyclic 2′ and 3′ phosphates of UMP have unique and well-resolved chemical shifts between 3 and 3.5 ppm. (Fig. 1c). When 2′,3′-cyclic UMP (cUMP) is incubated with yUsb1, a new peak at 3.4 ppm is formed that corresponds precisely to the chemical shift of $^{31}$P in a 3′ UMP standard (Fig. 1c)[26]. Additionally, the H6 resonance of uracil is well documented to be highly sensitive to the position of the phosphate at the 2′ vs. 3′ position[26, 27] and further confirms that the product is a 3′ phosphate (Supplementary Fig. 2a). Finally, two-dimensional $^{1}$H–$^{31}$P heteronuclear multiple bond correlation (HMBC) and $^{1}$H–$^{1}$H correlation spectroscopy (COSY) spectra unambiguously show that the product is a 3′ phosphate (Supplementary Fig. 2b, c). These data demonstrate that yUsb1 has CPDase activity that catalyzes the formation of a terminal 3′ phosphate, unlike metazoan Usb1, which lacks CPDase activity altogether[10, 11].

To determine how yUsb1 can occasionally remove more than one nucleotide, we tested its ability to process RNA substrates with different 3′ ends. yUsb1 is less efficient on 2′ phosphate-terminated substrates and is inactive on RNAs with a terminal 3′ phosphate (Fig. 1d). Substrates terminating in a 2′,3′-cyclic phosphate are processed by Usb1 with slower kinetics and to a lesser extent (Fig. 1d, lanes 6–8). These data demonstrate that yUsb1 is incapable of further processing its dominant 3′ phosphate product and that successive (*n-2* and *n-3*) products are likely formed from a cyclic phosphate intermediate that is reprocessed prior to second step CPDase chemistry. The amount of slower mobility *n-1* cyclic phosphate product is consistent with the relative kinetics of *n-2* product formation (Fig. 1b). The presence of both exonuclease and CPDase activity, along with inactivity on 3′ phosphate-terminated substrates, reveals an elegant mechanism for ensuring that yUsb1 does not overprocess and degrade U6 RNA.

**The conserved architecture and active site of Usb1**. We determined the crystal structure of the catalytic domain of yUsb1 (amino acids (a.a.) 71–290) to 1.8 Å resolution (Fig. 2a and Table 1). The enzyme exhibits a typical 2H phosphodiesterase fold[28] with an active site containing two H-X-S motifs (Fig. 2b). yUsb1 crystallized in the presence of 2 M ammonium sulfate, and the resulting structure contained a sulfate ion coordinated within the active site which likely mimics the binding mode of the scissile phosphate (Fig. 2b and Supplementary Fig. 3). NE2 of H231 is 2.8 Å from the sulfate ion, while H133 is farther away (4 Å). Residues 107–115 could not be modeled in our structure and are presumed to be disordered. Despite low sequence identity (<20%), the structure of yUsb1 is strikingly homologous to that of human Usb1[11], with nearly superimposable active sites (root mean square deviation for H/S in active site=0.3 Å).

**Structure of human Usb1 with a substrate analog**. To understand how Usb1 recognizes RNA and catalyzes 3′ end processing, we sought to determine the co-crystal structure of Usb1 with a substrate analog. We obtained a structure of truncated hUsb1 (a.a. 78–265) with 5′ UMP at 1.4 Å resolution (Fig. 2c), using the same crystal form as Hilcenko et al.[11], and incorporation of ligand via soaking of 10 mM 5′ UMP at pH ~6.5. In the structure of the apoenzyme at pH 5.6 (Protein Data Bank (PDB) ID 4H7W), the active site histidine H120 appears in two conformations, with one conformation tilted away from the active site and a proximal conformation stabilized by the coordination of a water molecule[11]. In the 5′ UMP-containing structure, H120 adopts the proximal conformation, and is 3.1 Å from an oxygen of the phosphate group in 5′ UMP (Fig. 2d). The 5′ UMP is held in place by hydrogen bonding interactions with active site residues H120, S122, S210 and H208, and a stacking interaction with Y202. The 5′ phosphate of 5′ UMP is positioned

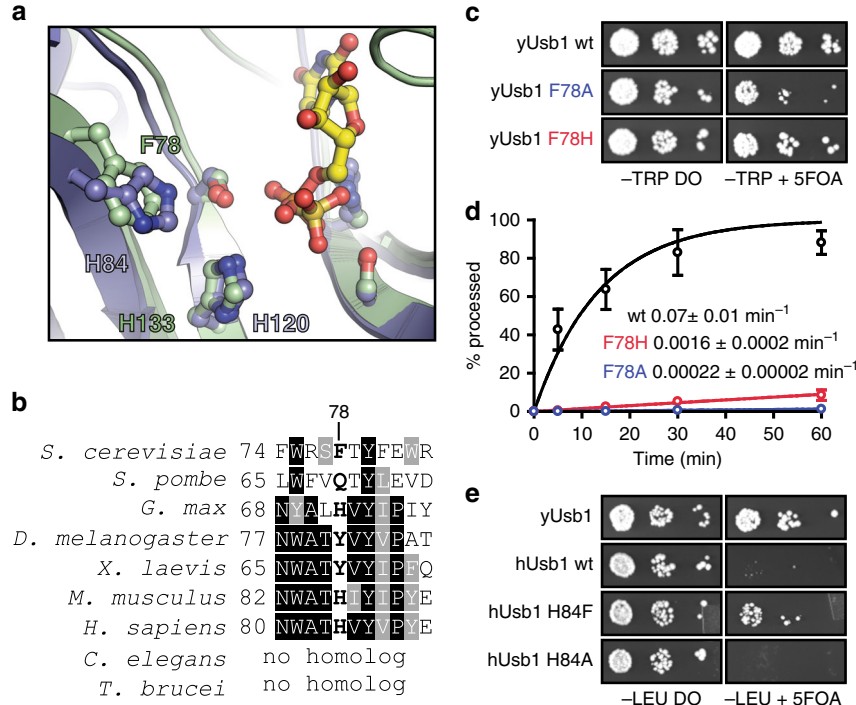

**Fig. 3** Residue F78 influences RNA processing by yUsb1. **a** Overlay of yUsb1 and hUsb1 active sites shows that yUsb1-F78 and hUsb1-H84 are positioned similarly adjacent to the active site. **b** Sequence alignment of the region surrounding yUsb1-F78/hUsb1-H84. Several residues (yUsb1 W75 and Y80) are perfectly or near-perfectly conserved, while conservation of position F78 is limited to aromaticity. **c** Usb1 mutants F78A and F78H can complement genomic deletion of *USB1* using plasmid shuffle and assaying for growth on media containing 5-fluoroorotic acid (5-FOA). Usb1-F78A results in a growth defect. **d** Quantification of the rate of Usb1 processing on a fluorescent substrate (as in Fig. 1) with different substitutions at position 78. Plotted data points represent the average of three technical replicates ± s.d. **e** Overexpression of yUsb1 (under control of the GPD promoter) complements deletion of *USB1*, but overexpression of hUsb1 results in an extreme slow growth phenotype. Overexpression of hUsb1-H84F partially rescues growth, while hUsb1-H84A does not

near the center of the active site, close to the position of the sulfate ion in the yeast active site which would correspond to the scissile phosphate that is involved in the first exonucleolytic step. We note that the hydrogen bonding interaction between H208 and the O5′ oxygen is positioned for general acid catalysis (Fig. 2d), consistent with an enzymatic mechanism proposed elsewhere[11]. We hypothesize that the binding mode of 5′ UMP in our structure of hUsb1 is analogous to how the last nucleotide of an RNA substrate would be coordinated within the active site.

**Comparison of human and yeast Usb1 enzymatic activities.** We further characterized the enzymatic activities of human and yeast Usb1 in order to better understand how they differ with respect to U6 processing. hUsb1 removes multiple nucleotides over the course of an hour (Fig. 2e), as observed previously[11]. Interestingly, hUsb1 is also strongly inhibited by a 3′ phosphate terminated RNA (Fig. 2f and ref. [11]). Thus, the mechanistic underpinning for the ability of hUsb1 to remove multiple nucleotides from U6 arises from its inability to catalyze "second step" cyclic phosphate ring opening. The structure of hUsb1 with 5′ UMP shows that RNAs with 3′ modifications would be sterically occluded by a loop spanning residues 161–167, suggesting that this region of the protein may contribute to substrate specificity.

Human Usb1 was less active in our assay conditions (pH 6.5) than in previous assays conducted at pH 8.0[11]. To reconcile this difference, we monitored the pH dependence of human Usb1 and found that it had a pH optimum of 7.5, in contrast to yeast Usb1, which exhibited a significantly different activity range and a pH optimum of 6.5 (Supplementary Fig. 4a, b). yUsb1 remains partially active even at pH 4, whereas hUsb1 is completely inactive. In contrast, at pH 10, yUsb1 is completely inactive,

whereas hUsb1 retains activity. This difference in pH optimum suggests that the active site histidine residues in yeast and hUsb1 have markedly different p$K_a$'s, likely due to different active site microenvironments. For example, hUsb1 has a histidine (H84) adjacent to the active site that can hydrogen bond to the active site serine S122[11]. In yeast, this position is a phenylalanine (F78) which cannot form an analogous hydrogen bond.

**A residue adjacent to the active site influences activity.** As yeast and human Usb1 exhibit remarkably similar H-X-S active sites, we inspected residues surrounding the active site for additional explanations as to how yeast and human Usb1 exhibit divergent enzymatic behaviors. We observed that Phe78 in yeast and His84 in human Usb1 are structurally homologous, suggesting a possible role for influencing water nucleophilicity during second step chemistry, or modulation of active site p$K_a$ (see above section) (Fig. 3a). With the exception of *S. pombe* Usb1, aromaticity (but not amino acid identity) is conserved at this position in other organisms (Fig. 3b). We therefore asked if mutations in yUsb1 at position 78 would have a phenotype in *S. cerevisiae*. Indeed, mutating F78 to alanine causes a slow growth phenotype in vivo (Fig. 3c) and a >300-fold reduction in the rate of processing in vitro (Fig. 3d). Surprisingly, F78H has no observable growth phenotype (Fig. 3c), yet has a 47-fold reduction in in vitro processing rate (Fig. 3d). Thus, significant reductions in Usb1 activity but not complete loss of the protein (Supplementary Table 1)[11] are tolerated by yeast.

Since mutation of F78 to histidine (as in hUsb1) supported yeast growth, we next asked if hUsb1 could complement deletion of *USB1* in yeast. When expressed from a high-copy plasmid using a GPD promoter, wild-type hUsb1 allows for an extreme slow growth phenotype, with small colonies visible only after

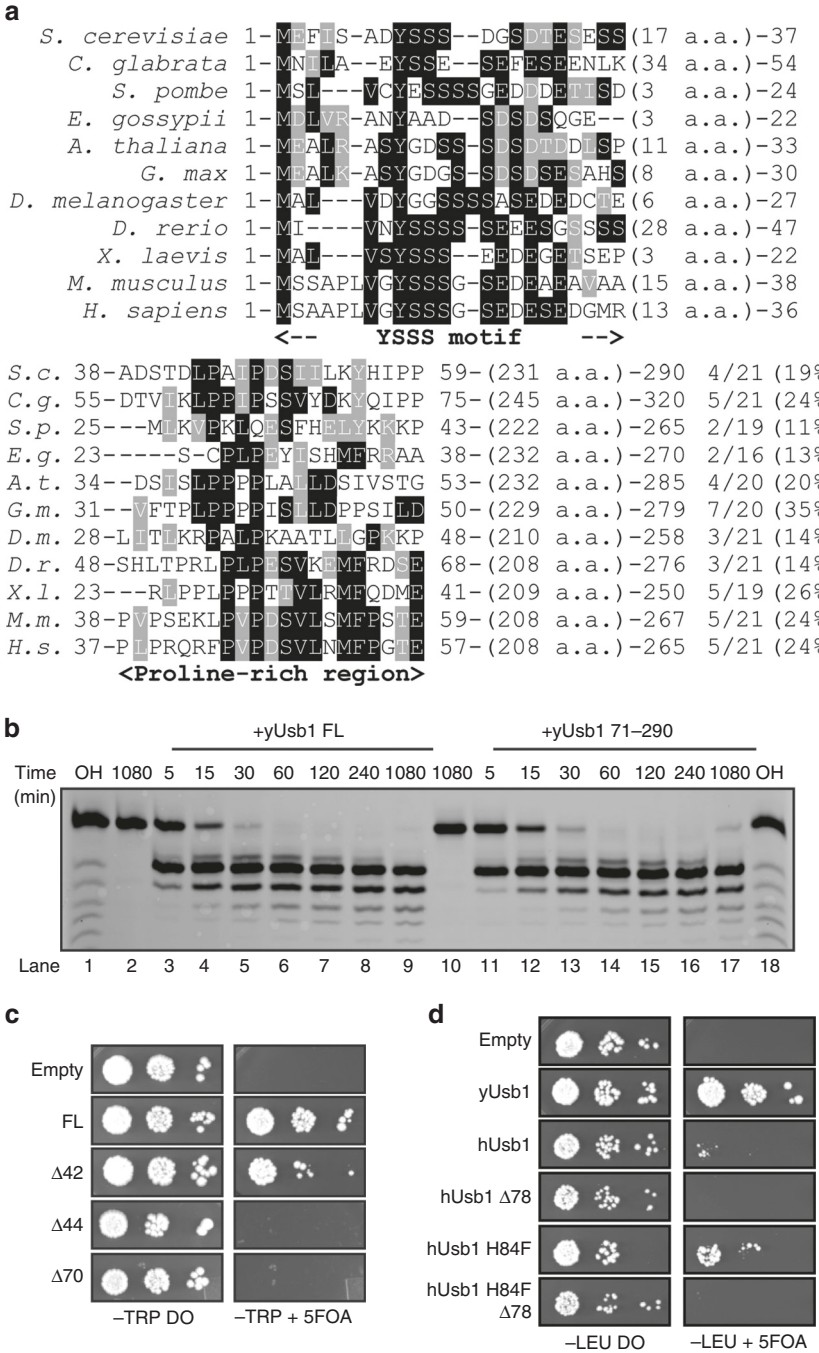

**Fig. 4** Usb1 catalytic activity alone is not sufficient for yeast viability. **a** Sequence alignment of N-terminal regions of Usb1. While the length of the N-terminus is poorly conserved (ranging from 46 to 103 residues in length), two regions of sequence conservation exist near the N-terminus (with a conserved Y-poly S-(D/E) motif) and in the center of the N-terminal region (the proline-rich region). Proline composition of this region is indicated to the right of the alignment. **b** Time course comparing the activities of full-length Usb1 (lanes 2–9) and the catalytic domain (residues 71–290) (lanes 10–17) over the course of 18 h shows similar rate and extent of processing for both proteins. Alkaline hydrolysis ladders (lanes 1 and 18) show the mobility of oligonucleotide products of different length. **c** N-terminal truncated forms of Usb1 show different capacity to support complementation of *USB1Δ*. Removing the first 42 residues supports growth, whereas removing 44 residues or more does not. **d** hUsb1-H84F does not complement *USB1Δ* if the homologous N-terminal region (Δ78) is deleted

>3 days (Fig. 3e). We substituted H84 in hUsb1 for the phenylalanine found in that position in yUsb1 and determined if that improved yeast proliferation. Indeed, hUsb1-H84F rescues yeast growth, whereas H84A does not, suggesting that H84F is a true gain-of-function mutant (Fig. 3e). Thus, mutations at this position greatly affect both in vitro processing rate and in vivo viability. Conversely, substitution of many perfectly conserved

residues outside of the active site (i.e., yUsb1 Y80) has no effect on yeast viability (Supplementary Table 1).

**N-terminus of Usb1 is essential for yeast viability.** Orthologs of Usb1 contain an N-terminal region that is predicted to be disordered[29, 30]. Surprisingly, the N-terminal region is

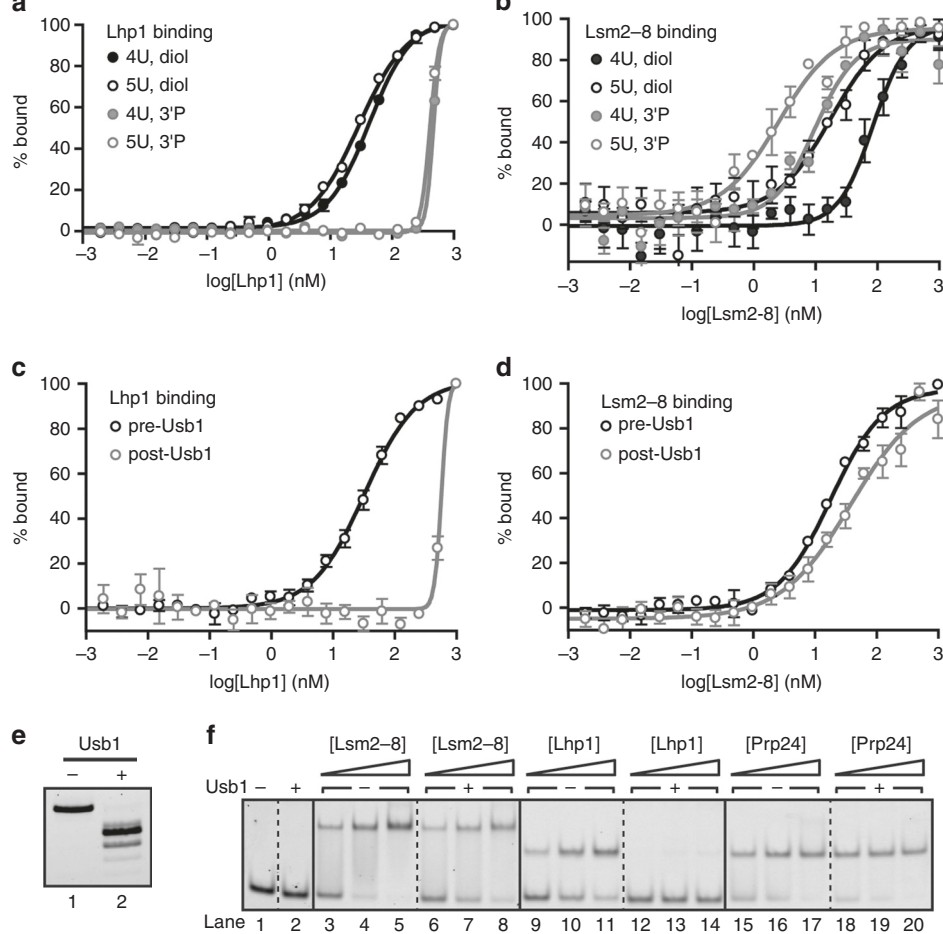

**Fig. 5** Usb1 processing influences RNP formation. **a** Fluorescence polarization binding data comparing Lhp1 binding to U6 95-112 with a *cis*-diol (*black, filled circles*), U6 95–112+1U with a *cis*-diol (*black, open circles*), U6 95–112 with a 3′ phosphate (*gray, filled circles*), and U6 95–112+1U with a 3′ phosphate (*gray, open circles*). Plotted data points represent the average of three technical replicates ± s.d. for (**a–d**). **b** Fluorescence polarization binding data comparing Lsm2–8 binding to U6 95–112 with a *cis*-diol (*black, filled circles*), U6 95–112+1U with a *cis*-diol (*black, open circles*), U6 95–112 with a 3′ phosphate (*gray, filled circles*) and U6 95–112+1U with a 3′ phosphate (*gray, open circles*). **c** Fluorescence polarization binding data comparing Lhp1 binding to U6 95–112+1U before (*black*) and after (*gray*) Usb1 processing. **d** Fluorescence polarization binding data comparing Lsm2–8 binding to U6 95–112+1U before (*black*) and after (*gray*) Usb1 processing. **e** Denaturing gel comparing RNAs used in **c**, **d**. **f** The affinities of Lsm2–8 and Lhp1 for full-length U6 are influenced by Usb1 processing. Native gel analysis comparing Lsm2–8, Lhp1 and Prp24 affinity for U6 RNA before and after treatment with Usb1. Usb1 processing does not change the mobility of U6 on a native gel (lanes 1 vs. 2). Lsm2–8 binds similarly before (lanes 3–5) and after (lanes 6–8) Usb1 processing. Lhp1 binding (lanes 9–11) is negligible after Usb1 processing (lanes 12–14). Prp24 binding is unchanged before (lanes 15–17) and after Usb1 processing (lanes 18–20)

more conserved across species than the catalytic domain (Fig. 4a). Although the length of the domain is highly variable, there are two regions that are conserved in sequence: a Y(S)$_N$(D/E) motif in the first 20 amino acids, and a proline-rich region in the center of the domain. The first serine-rich motif could be a site for post-translational modification, while the second, proline-rich region could be important for a protein–protein interaction, stability or expression of Usb1, as proline-rich regions are important for many protein–protein interactions[31]. In this region of ~20 amino acids, ~20% of the residues (and up to 35% in *Glycine max* Usb1) are prolines (Fig. 4a). We asked what role, if any, the N-terminal domain plays in catalysis or function of yUsb1 in vivo.

We first compared the in vitro activity of the catalytic domain of yUsb1 (a.a. 71–290) to that of full-length yUsb1. The N-terminal region (residues 1–70) has no observable effect in our in vitro exonuclease assay as the rate and extent of processing for full-length and the catalytic domain of yUsb1 are indistinguishable (Fig. 4b). Hilcenko et al.[11] previously reported that an N-terminal deletion of yUsb1 (Usb1 77–290) failed to complement *USB1Δ* in yeast. Our crystal structure reveals that residues 73–76 form the

start of a β-strand and make several intramolecular contacts that are likely important for folding. However, we find that yUsb1 71–290 (Δ70) is also insufficient to support yeast growth (Fig. 4c). This result is surprising, because the catalytic domain is fully active in vitro and because the N-terminal region (residues 1–70) is predicted to be largely disordered. By making successive truncations (Supplementary Table 1), we found that residues 43–290 are essential for viability (Fig. 4c). This essential fragment correlates with the start of the proline-rich region (Fig. 4a). The inviability of Δ70 or Δ50 alleles of *USB1* could not be rescued by inclusion of an N-terminal SV40 nuclear localization signal (Supplementary Table 1). This suggests that the role of the N-terminus is more complex than controlling subcellular localization of Usb1.

When full-length and truncations of yUsb1 are expressed under control of a GPD promoter to achieve greater expression levels, we find that more extensive truncations of the N-terminus can complement *USB1Δ*; however, the Δ70 allele is still lethal (Supplementary Fig. 5a). Western blot analysis shows that truncation of the N-terminal region of yUsb1 results in reduced levels of Usb1 (Supplementary Fig. 5b). Deletion of the first 42 or 44 residues results

**Table 2 Lsm2–8 and Lhp1 binding parameters for U6 RNA with different 3′ ends**

| RNA | Lhp1 | | | | Lsm2–8 | | |
|---|---|---|---|---|---|---|---|
| U6 3′ tail | $K_d$ (nM) | Hill coefficient | $R^2$ | | $K_d$ (nM) | Hill coefficient | $R^2$ |
| U6 95–112 **3′OH** | 43 ± 1 | 1.27 ± 0.05 | 0.99 | | 85 ± 1 | 1.6 ± 0.3 | 0.89 |
| U6 95–112+1U **3′OH** | 29 ± 1 | 1.11 ± 0.04 | 0.99 | | 17 ± 1 | 1.0 ± 0.2 | 0.87 |
| U6 95–112 **3′P** | 467 ± 1 | 8 ± 2* | 0.99 | | 10 ± 1 | 1.3 ± 0.3 | 0.92 |
| U6 95–112+1U **3′P** | 411 ± 1 | 5.9 ± 0.5* | 0.99 | | 3 ± 1 | 1.0 ± 0.2 | 0.90 |
| **Full-length U6** | $K_d$ (nM) | $B_{max}$ (% bound) | $R^2$ | | $K_d$ (nM) | $B_{max}$ (% bound) | $R^2$ |
| U6 1–112 **3′OH** | 18 ± 1 | 75 ± 1 | 0.97 | | 210 ± 90 | 100 ± 20 | 0.89 |
| U6 1–112+1U **3′OH** | 12 ± 1 | 75 ± 1 | 0.98 | | 30 ± 3 | 100 ± 3 | 0.98 |
| U6 1–112+1U**>p** | n.d. | n.d. | n.d. | | 70 ± 10 | 100 ± 7 | 0.95 |

*The high Hill coefficient for Lhp1 binding to RNAs with 3′ phosphate ends likely indicates nonspecific aggregation. Bold text in the RNA column indicates 3′ end modification

in a significantly reduced signal from Usb1, and deletion of the first 70 amino acids results in an undetectable amount of Usb1, suggesting that the N-terminus of Usb1 plays a role in protein stability.

Taking advantage of the viable hUsb1-H84F mutation, we discovered that the N-terminal region of hUsb1 (residues 1–78) is also required for growth (Fig. 4d). This result suggests that the N-terminal domain may possess a conserved function. How the N-terminal domain influences stability, whether through an as-yet unidentified interaction or through another mechanism, remains to be determined.

**Usb1 processing directly controls formation of U6 RNPs**. We next investigated the impact of U6 3′ end processing on the affinity of U6 snRNA 3′ end binding proteins. We tested binding of Lhp1 and Lsm2–8 using a fragment of the U6 3′ end (U6 95–112) with either a *cis*-diol or a 3′ phosphate. As expected, Lhp1 greatly prefers to bind a *cis*-diol, with a $K_d$ of 43 nM, with essentially no specific binding to a 3′ phosphate (Fig. 5a and Table 2)[32]. In contrast, Lsm2–8 can bind both a *cis*-diol and a 3′ phosphate, but preferentially binds a 3′ phosphate ($K_d$ of 85 vs. 10 nM for U6 95–112 with a *cis*-diol or 3′ phosphate, respectively) (Fig. 5b and Table 2). Thus, exchange of a *cis*-diol for a 3′ phosphate eliminates Lhp1 binding[32] and improves Lsm2–8 binding affinity for U6. In humans, Lsm2–8 has been reported to preferentially bind a cyclic phosphate over a *cis*-diol[15], suggesting that the Lsm2–8 complex in different organisms has evolved to bind the product of Usb1 processing.

Yeast U6 snRNA possesses an oligouridylate tail of heterogeneous length[33] and we therefore compared the affinity of both Lhp1 and Lsm2–8 for U6 95–112 oligonucleotides with or without an additional terminal uridine. As expected, Lhp1 binds U6 95–112 with similar affinity to U6 95–112+1U (Fig. 5a and Table 2). In contrast, we find that Lsm2–8 binds U6 95–112+1U >4-fold tighter than U6 95–112 regardless of the 3′ end modification (Fig. 5b and Table 2). An extra uridine in the context of full-length U6 RNA also enhances Lsm2–8 binding (Supplementary Fig. 6 and Table 2). Thus, of the RNAs tested, Lsm2–8 has the highest affinity for U6 with 5 uridine residues and a 3′ phosphate modification.

Next, we directly monitored the effect of Usb1 processing on Lhp1 and Lsm2–8 binding to oligonucleotides and full-length U6 RNA. Usb1 processing of oligonucleotides produces binding profiles for Lhp1 that mirror the results with chemically defined 3′ ends (Fig. 5c, d), except for the fact that additional shorter products are also formed (Figs. 1b and 5e) that reduce Lsm2–8 affinity, as Lsm2–8 does not efficiently bind an RNA with only three terminal uridines[34]. The effect of Usb1 processing on U6-binding proteins also extends to full-length U6. Using full-

length U6 with an additional uridine at the 3′ end (U6 1–112 +1U), we tested the affinity of Lhp1, Lsm2–8 and Prp24 before and after Usb1 processing via native gel shift (Fig. 5f). Lsm2–8 can bind both before and after Usb1 treatment (Fig. 5f, lanes 3–5 vs. 6–8), but binding is slightly reduced due to the reduction in the U-tail length (Fig. 5b). Lhp1 binding is most sensitive to Usb1 processing, with tight binding before processing and virtually no binding after (Fig. 5f, lanes 9–11 vs. 12–14). Prp24 affinity is unchanged (Fig. 5f, lanes 15–17 vs. 18–20), as expected from the structure of the Prp24-U6 RNA complex[35]. These data clearly demonstrate that the modification at the 3′ end is the most important determinant for Lhp1 and Lsm2–8 binding and that processing by Usb1 controls the transition from immature (Lhp1-bound) to mature (Lsm2–8 bound) U6.

**Ordered binding in the U6 snRNP assembly pathway**. It is well established that Lhp1 binds to newly synthesized U6 RNA[36, 37]. Previous work on the subsequent steps in the U6 lifecycle has established that Prp24 and Lsm2–8 cooperatively bind U6 RNA[15] and that Prp24 and Lsm2–8 directly interact via the C-terminus of Prp24[18]. However, the effect of Prp24 on binding of Lhp1 has not been investigated. We tested co-binding of Prp24 and either Lhp1 or Lsm2–8 (Fig. 6a). Prp24 binds U6 with a $K_d$ that is 10-fold lower than Lhp1 (Didychuk et al.[17]; Fig. 6a, lanes 2–6 and 7–11). When U6 RNA is incubated with equimolar amounts of Prp24 and Lhp1, it is preferentially bound by Prp24 and a ternary complex is not visible (lanes 12–16). For example, Lhp1 alone is mostly bound to U6 at 160 nM, but in the presence of Prp24 no Lhp1 binding is observed until 640 nM (Fig. 6a, compare lanes 4 and 16). Formation of a U6–Lhp1–Prp24 ternary complex occurs only at high concentrations (lane 16). These data suggest that Prp24 binding is anti-cooperative with binding of Lhp1. In contrast, Prp24 and Lsm2–8 bind cooperatively and efficiently form a ternary complex (Fig. 6a, lanes 22–26). While Lsm2–8 binds U6 with a *cis*-diol relatively weakly by itself (Fig. 6a, lanes 17–21), inclusion of Prp24 strongly promotes formation of a ternary complex (Fig. 6a, lanes 22–26) even at the lowest concentration tested and despite the lack of a phosphoryl group on U6 to promote Lsm2–8 binding.

The apparent negative cooperativity of Prp24 and Lhp1 binding is intriguing, since known Prp24 and Lhp1 binding sites on U6 RNA are presumed to be non-overlapping[35, 38]. We tested the affinity of Prp24 or Lhp1 for U6 pre-saturated with the opposing binding partner (Fig. 6b). When U6 is pre-bound by Lhp1, Prp24 displaces U6 from U6–Lhp1 (Fig. 6b, lanes 9–14). Consistent with these data, U6 pre-bound by Prp24 is not released from U6–Prp24 by the addition of excess Lhp1 (Fig. 6b, lanes 23–28). In both cases, ternary complex formation is

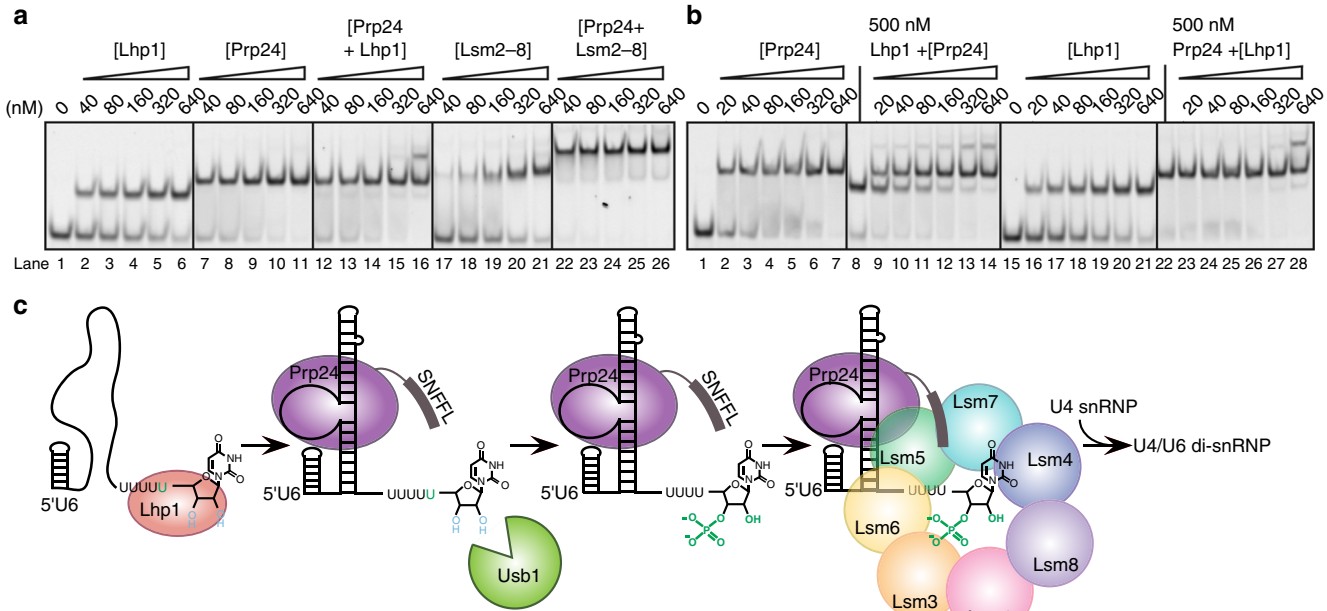

**Fig. 6** The U6 snRNP assembly pathway. **a** Native gel analysis of U6 binding partners. Lhp1 and Prp24 bind U6 1–112 with a *cis*-diol tightly (lanes 2–6 and 7–11). Inclusion of equimolar amounts of Lhp1 and Prp24 does not promote formation of a ternary complex except at the highest concentration (lanes 12–16). In contrast, Lsm2–8 binds U6 relatively weakly (17–21), but upon inclusion of Prp24 (lanes 22–26), Lsm2–8 efficiently forms a co-complex of U6, Lsm2–8 and Prp24. **b** Prp24 binds naked U6 1–112 with a *cis*-diol (lanes 2–7) and U6 pre-saturated with Lhp1 (lanes 9–14) tightly. Prp24 abstracts U6 from U6-Lhp1 much more efficiently than it forms a U6/Prp24/Lhp1 complex. Lhp1 binds naked U6 (lane 21), but cannot bind or release U6 from pre-formed U6-Prp24. **c** Model of U6 snRNP assembly. U6 is synthesized by RNA polymerase III and initially bound by Lhp1. Binding of Prp24 weakens Lhp1 affinity for the 3′ tail of U6, allowing Usb1 to remove a uridine and leave a 3′ phosphate modified tail. Lsm2–8 recognizes the 3′ tail of U6 and interacts with Prp24 to form the U6 snRNP, which can then be assembled into the spliceosome via the U4/U6 di-snRNP

inefficient and occurs only at high concentration, suggesting that Prp24 and Lhp1 are near-mutually exclusive for binding. From these data we propose a model for U6 snRNP assembly (Fig. 6c). Synthesis by Pol III followed by binding of Lhp1 protects nascent U6 from 3′ exonucleases. Binding of U6 by Prp24 is anti-cooperative with binding of Lhp1, allowing the freed 3′ tail that can then be recognized by Usb1 to produce a 3′ phosphate. The presence of a 3′ phosphate group prevents re-binding of Lhp1 and allows for recruitment of Lsm2–8, whose interaction is stabilized by recognition of the 3′ phosphate (Fig. 5) and by interactions with the Prp24 C-terminal SNFFL box[15, 18]. The U6 snRNP can then join with the U4 snRNP[17] for efficient incorporation into the tri-snRNP and the spliceosome.

## Discussion

Members of the 2H phosphodiesterase superfamily are found in viruses, Archaea, bacteria and eukaryotes, and are involved in diverse pathways of RNA metabolism. 2H phosphodiesterase enzymes are highly divergent in sequence outside of the active site H-X-S/T motif[28], yet structurally homologous. A DALI search[39] (Supplementary Table 2) reveals that the closest structural homologs of yUsb1 (after hUsb1) include putative 2′–5′ ligases[40, 41], 2′,3′-cyclic nucleotide 3′-phosphodiesterases (such as the well-characterized enzyme ThpR)[42, 43], 2′–5′ phosphodiesterases[44, 45] and several proteins of unknown activity[46, 47]. Thus, the fold of Usb1 is more reminiscent of a 2′,3′-cyclic phosphodiesterase than an exoribonuclease, and the closest homologs tend to open cyclic phosphates to produce a 2′ phosphate instead of a 3′ phosphate. Here, we have demonstrated that two members of the 2H phosphodiesterase superfamily, yUsb1 and hUsb1, are mechanistically distinct, yet are involved in the same step of U6 biogenesis in their respective organisms.

Yeast Usb1 possesses two RNA processing activities (3′–5′ exonuclease activity and 2′,3′-cyclic nucleotide 2′ phosphodiesterase activity), while hUsb1 lacks CPDase activity. Our results suggest that while the active sites of the 2H phosphodiesterase family are highly similar, residues immediately outside of the active site likely play a previously unappreciated role in RNA recognition and catalysis. Indeed, three highly structurally homologous enzymes—ThpR, yUsb1 and hUsb1—possess virtually superimposable H-X-S/T active sites, but have unique 2′,3′-cyclic phosphodiesterase activities (catalyzing formation of 2′ phosphate[43], 3′ phosphate (this work) and 2′,3′-cyclic phosphate[11] products, respectively). Enzyme active sites are dynamic and highly sensitive to sub-angstrom scale differences, making prediction of structure–activity relationships difficult. Further study of other 2H phosphodiesterases may reveal additional diversity of substrate specificities and activities for this superfamily.

The 3′ phosphate terminus generated by yUsb1 inhibits subsequent exonucleolytic processing, whereas hUsb1 leaves a 2′,3′-cyclic phosphate terminus which is a substrate for an additional exonucleolytic step, resulting in successive trimming of U6[11]. Humans also possess a counteracting TUTase that extends the 3′ tail of U6[48, 49]. *S. cerevisiae* has no identifiable TUTase and thus the autoinhibition of yUsb1 by its own product is necessary to prevent overprocessing of U6. The combined dual exonuclease and CPDase activity of Usb1 accomplishes three important functions to initiate U6 snRNP biogenesis: it prevents further 3′ end processing, improves Lsm2–8 affinity for U6 RNA and, perhaps more importantly, prevents Lhp1 from rebinding. A 3′ phosphate might encounter steric clash with nearby loop residues of Usb1.

The evolutionary divergence in Lsm2–8 binding affinity appears to have co-evolved with the activity of Usb1. Yeast Lsm2–8 tightly

binds RNAs with a 3′ phosphate, while human Lsm2–8 prefers to bind RNAs with a 2′,3′-cyclic phosphate[15]. Interestingly, yeast Lsm2–8 appears to bind a 2′,3′-cyclic phosphate even worse than an unmodified *cis*-diol or a noncyclic phosphate. Therefore, yeast Lsm2–8 and human Lsm2–8 significantly differ in their modes of 3′ end recognition and apparently have evolved to bind the product of their cognate Usb1. Along with this observation, we also demonstrate that Lsm2–8 binds the oligonucleotide U6 95–112 with a *cis*-diol >25-fold worse than a U6 95–112+1U oligonucleotide with 3′ phosphate (Table 2). This model substrate, with five terminal uridines and a phosphate end, is very similar to the major mature form of human U6, which contains five terminal uridines and a cyclic phosphate end[13]. While there is a co-crystal structure of yeast Lsm2–8 with RNA, it was crystallized with an oligonucleotide containing a *cis*-diol rather than a 3′ phosphate and containing four terminal uridines instead of five, making it difficult to identify the crucial interactions that promote correct 3′ end recognition[34]. Our data indicate that both terminal 3′ phosphate and the length of the oligouridylate tail are important binding determinants for Lsm2–8. It would be interesting to correlate evolutionary changes in Usb1 and Lsm2–8 with the binding mode of the Lsm2–8 complex from yeast and humans to the appropriate biological 3′ end of U6.

Studies in fission yeast and human cell lines reveal that U6 (and U6atac in humans) is the major substrate of Usb1[13]. This, along with the observation that overexpression of U6 can rescue loss of Usb1 in *S. cerevisiae*[11], suggests that Usb1 is an exonuclease that is highly specific for U6 RNA. How is such stunning specificity accomplished within the cell, where many other RNA polymerase III transcripts also terminate in a polyuridine stretch with a 2′,3′-*cis*-diol? Usb1 is similarly active on short RNA oligonucleotides and on full-length U6 snRNA, excluding specific recognition of U6 secondary structure by Usb1. We hypothesize that specificity stems from the lack of accessible substrates for Usb1 due to binding of Lhp1 on the majority of $(U)_n$ *cis*-diol-containing RNAs. Binding of Prp24, however, is highly specific for the asymmetric bulge in U6 snRNA[17, 35]. We have demonstrated that Prp24 binding to U6 lowers the apparent affinity of Lhp1 (Fig. 6a, b); this, in turn, would allow for Usb1 to modify the free 3′ end and thereby promote binding of Lsm2–8. The mechanism for handoff of U6 from Lhp1 to Prp24 is currently unknown, but may be caused by partially overlapping binding sites of Prp24 and Lhp1, occlusion of Lhp1 via electrostatic repulsion or induction of RNA folding that is unfavorable for Lhp1 association. Future studies aimed at determining the origin of the observed anti-cooperative binding could reveal fundamental principles of ordered, multi-step pathways of RNP assembly.

This handoff of U6 RNA, from Lhp1 to Prp24 to Prp24/Lsm2–8, ensures that full-length and properly 3′-end-modified U6 is guarded from exonucleases in the cell. Presence of a terminal 3′ phosphate may partially protect U6 RNA from degradation by the exosome, as Rrp6 is inactive on a 3′ phosphate-terminated RNA[50], but Rrp44/Dis3 is active on such substrates[51]. This work illustrates how Usb1 processing initiates U6 snRNP formation through a series of protein–RNA interactions that have evolved to protect U6 snRNA and chaperone it into the active site of the spliceosome.

## Methods

**Protein expression and purification.** The Usb1 coding sequence was PCR amplified from *S. cerevisiae* genomic DNA with primers to introduce flanking BamHI and XhoI sites and subcloned into a pET28b plasmid, which encodes an N-terminal hexahistidine tag followed by a TEV cleavage site. The resulting cloning scar was removed using the inverse PCR method with Phusion DNA polymerase (New England Biolabs). All primer sequences are listed in Supplementary Table 3. PCR products were DpnI treated, ligated using T4 PNK and T4 DNA ligase and transformed into *Escherichia coli* NEB 5α-competent cells (New England Biolabs). Expression plasmids contained Usb1 residues 1–290

(full-length) or 71–290 (catalytic domain). Mutants of these plasmids were obtained using inverse PCR as described above. Resulting clones were expressed in *E. coli* BL21 STAR (DE3) pLysS cells (Invitrogen) in LB at 37 °C with late-log phase induction by addition of 1 mM IPTG and subsequent growth for 3 h at 37 °C (for Usb1 71–290) or 16 °C (for Usb1 1–290 and all other proteins used in this study). Cells were collected by centrifugation, resuspended in IMAC buffer (500 mM NaCl, 50 mM HEPES acid, 50 mM sodium HEPES base, 15 mM imidazole base, 10% v/v glycerol, 1 mM TCEP-HCl) supplemented with DNase I, lysozyme and protease inhibitors (EMD Millipore) and lysed via sonication. Insoluble material was removed by centrifugation. Usb1 was purified by Ni-NTA agarose chromatography by step elution with IMAC buffer containing 500 mM imidazole. The eluate was dialyzed at 4 °C overnight with 1 mg TEV protease into IEX-HEPES buffer (100 mM NaCl, 10 mM HEPES acid, 10 mM sodium HEPES base, 10% v/v glycerol, 1 mM TCEP-HCl, pH ~ 7.0) for Usb1 71–290 or IEX-bis-tris buffer (100 mM NaCl, 20 mM bis-tris, 10 mM HCl, 10% v/v glycerol, 1 mM TCEP-HCl, pH ~ 6.2) for Usb1 1–290. Precipitated protein was removed, then protein was further purified via cation-exchange chromatography (HiTrap S, GE Healthcare) in IEX buffer with gradient elution against IEX buffer containing 2 M NaCl.

The coding sequence for *Homo sapiens* Usb1 was codon optimized for expression in *E. coli*. The sequence of the synthetic hUsb1 gene is shown in Supplementary Table 4. For functional assays, this sequence was cloned via NdeI and BamHI restriction cloning into a modified pET3a plasmid (Novagen) containing an N-terminal octahistidine tag, glutathione-S-transferase (GST), and a TEV (tobacco etch virus) cleavage site. Human Usb1 (residues 79–265) used in crystallography experiments was similarly cloned into a modified pET3a plasmid (Novagen) containing an N-terminal octahistidine tag, MBP, and a TEV cleavage site. Protein was expressed and purified as described above using Ni-NTA agarose chromatography and cation-exchange chromatography with IEX-HEPES buffer.

The coding sequence for *S. cerevisiae* Lhp1 protein (residues 1–275) was codon optimized for expression in *E. coli*. The sequence of the synthetic Lhp1 gene is shown in Supplementary Table 4. A pET3a plasmid (Novagen) was modified using inverse PCR as described above to encode an N-terminal octahistidine tag followed by a TEV cleavage site. The coding sequence for Lhp1 was cloned into this plasmid using the NdeI and BamHI sites. Protein was expressed and purified as described above using Ni-NTA agarose chromatography and cation-exchange chromatography with IEX-HEPES buffer.

The Prp24 coding sequence was PCR amplified from *S. cerevisiae* genomic DNA with primers to introduce flanking NdeI and XhoI sites and subcloned into a pET21b plasmid, which encodes a C-terminal hexahistidine tag. Prp24 was expressed as described above in Terrific Broth. Prp24 was purified by Ni-NTA agarose chromatography as described above using Ni-NTA agarose chromatography with IMAC buffer containing 50 mM imidazole, dialyzed without TEV protease and purified via heparin chromatography (HiTrap Heparin, GE Healthcare) in IEX-HEPES as described above with the addition of 1 mM sodium azide.

*S. cerevisiae* Lsm2–8 lacking the C-terminus of Lsm4 was expressed from pQLink-Lsm2–8[34] in *E. coli* BL21 STAR (DE3) pLysS cells in Terrific Broth as described above. Lsm2–8 was first purified via Ni-NTA agarose chromatography via a TEV-labile polyhistidine tag on Lsm8 in IMAC buffer containing 50 mM imidazole base, then dialyzed overnight into IEX-HEPES buffer. After removal of precipitated protein, Lsm2–8 was purified via a TEV-labile GST tag on Lsm6 and glutathione agarose chromatography with step elution in IEX-HEPES supplemented with 10 mM reduced glutathione, 50 mM HEPES acid and 50 mM sodium HEPES base. The polyhistidine and GST tags were removed by the addition of 1 mg TEV protease during overnight dialysis at room temperature into IEX-HEPES. Lsm2–8 was then purified via anion-exchange chromatography (HiTrap Q, GE Healthcare) with gradient elution in IEX-HEPES supplemented with 2 M NaCl. Lsm2–8 was then diluted fivefold against IEX-bis-tris containing 1 mM sodium azide and further purified via heparin chromatography (HiTrap Heparin, GE Healthcare) in IEX-bis-tris containing 1 mM sodium azide with gradient elution in buffer supplemented with 2 M NaCl.

A truncated variant of AtRNL containing only the kinase and 2′,3′-cyclic phosphate 3′-phosphodiesterase domains (residues 677–1104) was modified from pET28-Smt3-AtRNL (1–1104) (a kind gift from Stewart Shuman[52]) via inverse PCR to remove the N-terminal ligase domain and install a TEV cleavage site after the octahistidine tag. Overexpression and purification was essentially as described above using Ni-NTA agarose chromatography with IMAC buffer with 50 mM imidazole and cation-exchange chromatography using IEX-HEPES. All protein samples were analyzed by sodium dodecyl sulfate–polyacrylamide gel electrophoresis to assess purity.

**Crystallization and structure determination.** Crystals of truncated yUsb1 (residues 71–290) were obtained by hanging drop vapor diffusion with 2 μL concentrated protein (9 mg mL⁻¹) and 2 μL of crystallization solution (0.1 M sodium acetate, pH 4.5, 2.0 M ammonium sulfate) with equilibration against 500 μL of crystallization solution at 20 °C. Crystals grew as needles of approximate dimensions 50 × 50 × 100 μm over 1–3 days. Crystals were cryoprotected by addition of 10 μL of a solution containing 15% v/v glycerol, 20% w/v PEG 20,000, 0.1 M

sodium acetate pH 4.6 to the crystallization drop. A heavy atom derivative was produced as above using a cryoprotectant solution that was also saturated with uranyl acetate and incubation for 1 min prior to freezing. Diffraction data were collected at 100 K on beamlines 21-ID-F or 24-ID-C at the Advanced Photon Source. Data were integrated using XDS[53]. Space group determination and scaling were performed in POINTLESS[54] and AIMLESS[55], respectively. Phenix.xtriage was used to assay potential twinning in the diffraction data[56]. Initial phases could not be determined using molecular replacement with hUsb1 78–265 (PDB 4H7W), and therefore phases were determined by the method of Single Isomorphous Replacement with Anomalous Scattering, using initial heavy atom site identification, map calculation and density modification in the SHELXC/D/E pipeline[57] as implemented in HKL2Map[58]. Automated model building was accomplished with RESOLVE[56, 59], with subsequent refinement via iterative rounds of manual model building in Coot[60] and automated refinement in PHENIX[56, 61].

Attempts to co-crystallize or soak in nucleotides into yUsb1 were unsuccessful, likely due to the stringent requirement for ammonium sulfate in the crystallization conditions. Crystals of truncated hUsb1 (residues 79–265) were grown by hanging drop vapor diffusion in 1.4 M sodium potassium phosphate pH 5.6 at 289 K and then transferring into a solution containing 20% w/v PEG 20,000, 20% v/v glycerol, 100 mM bis-tris base, 50 mM HCl and 10 mM disodium 5′ UMP and allowing to incubate overnight. Data were collected on beamline 24-ID-C at the Advanced Photon Source and processed as above. Initial phases were determined by molecular replacement using Phaser[62] with PDB entry 4H7W[11], and refinement accomplished by the iterative process above. Data collection and refinement statistics are given in Table 1. All figures were generated with PyMOL (http://www.pymol.org).

**RNA production**. RNA oligonucleotides containing *S. cerevisiae* U6 nucleotides 95–112 with one, three, or six additional 3′ uridine nucleotides and containing a 5′ 6-carboxyfluorescein (6-FAM) moiety and a 2′,3′-*cis*-diol or phosphate were purchased from Integrated DNA Technologies. Sequences of all synthetic RNA oligonucleotides are listed in Supplementary Table 5. RNAs were purified via 20% 19:1 acrylamide:bis-acrylamide polyacrylamide gel electrophoresis (PAGE) containing 8 M urea, 89 mM Tris borate and 2 mM EDTA and extracted into 0.3 M NaOAc/1 mM EDTA. Oligonucleotides were then further purified via anion exchange (HiTrap Q column, GE Healthcare Life Sciences) using 100 mM NaCl, 20 mM bis-tris pH 6.5, 1 mM EDTA and elution with 2 M NaCl, 20 mM bis-tris pH 6.5 and 1 mM EDTA. RNAs were concentrated and stored in 50 mM NaCl, 20 mM bis-tris, pH 6.5, 1 mM EDTA.

To obtain an RNA with a *cis*-diol, cyclic phosphate, 2′ phosphate, or 3′ phosphate, a longer RNA oligomer comprising *S. cerevisiae* U6 nucleotides 84–112 with three additional 3′ uridine nucleotides was produced by splinted ligation. U6 95–112 (UUU) was produced via in vitro transcription from a plasmid containing a 5′ hammerhead ribozyme (HH) and a 3′ hepatitis delta virus ribozyme (HDV). This plasmid was generated via inverse PCR by modification of a plasmid containing a 5′ hammerhead ribozyme, full-length U6 1–112, and a 3′ HDV ribozyme (a kind gift from Kiyoshi Nagai). U6 95–112 (UUU), which contained a 5′ OH group and 2′,3′-cyclic phosphate group due to HH/HDV cleavage, was 5′ phosphorylated with ATP by either T4 PNK, T4 PNK (3′ phosphatase minus) or truncated AtRNL. The RNA was then phenol/chloroform/isoamyl alcohol extracted to remove enzyme, then ligated to 5′ 6-FAM U6 84–94 (Integrated DNA Technologies) with T4 RNA ligase 2 at 37 °C for 2 h using a DNA splint that was complementary to the entire RNA product. The ratio of U6 95–112 (UUU)/FAM U6 84–94/DNA splint was 1:1.5:2. The ligation product was then gel purified, extracted and purified via ion exchange as described above. Fluorescent U6 84–112 (UUU) with a 3′ phosphate was prepared synthetically by Integrated DNA Technologies.

Full-length Cy3-labeled U6 (nucleotides 1–112) and full-length U6 containing one additional 3′ uridine nucleotide (U6 1–112+1U) were produced by splinted ligation as described above using a 5–Cy3 U6 1–12 RNA oligonucleotide (Integrated DNA Technologies) and U6 13–112 that was produced via in vitro transcription from a modified plasmid containing a 5′ hammerhead ribozyme and a 3′ HDV ribozyme (a kind gift from Kiyoshi Nagai).

**NMR spectroscopy**. Samples of 1 mM uridine-2′,3′-cyclic monophosphate sodium salt (2′,3′-cUMP; Santa Cruz Biotechnology) in 50 mM NaCl, 20 mM bis-tris base, 10 mM HCl, 30 μM DSS and >95% D$_2$O incubated with either buffer (100 mM NaCl, 10 mM HEPES base, 10 mM HEPES acid, 10% v/v glycerol, 1 mM TCEP pH~7.2), 8 μM AtRNL or 75 μM Usb1 71–290 overnight at room temperature. All spectra were obtained on a BrukerAvance III 600MHz spectrometer with a 5 mm $^1$H ($^{13}$C/$^{15}$N/$^{31}$P) cryogenic probe at the National Magnetic Resonance Facility at Madison (NMRFAM). $^1$H chemical shifts were directly referenced to DSS and $^{31}$P chemical shifts were indirectly referenced to DSS. The 3′ phosphate left after Usb1 treatment was assigned based on chemical shift in 1D $^{31}$P spectra and comparison to 3′ UMP and 2′,3′-cUMP controls. Presence of a 3′ phosphate was further confirmed via $^1$H–$^{31}$P HMBC and $^1$H–$^1$H COSY experiments.

**Statistical analysis**. Unless otherwise noted, all activity and binding assays were carried out in triplicate. The resulting data points were averaged before being used to calculate rate or binding constants. Plotted data points represent these averages ± s.d. Errors in rate or binding constants represent the errors generated by the fits to these data using GraphPad Prism 4 software.

**Exoribonuclease assays**. Exoribonuclease assays were performed in 10 μL reactions by mixing equal volumes of Usb1 (1 μM) in 100 mM KCl, 20 mM bis-tris, 10 mM HCl, 1 mM TCEP-HCl, 20% w/v sucrose, 0.01% v/v Triton X-100, 0.2 mg mL$^{-1}$ bovine serum albumin (BSA) and RNA substrate (200 nM) in 100 mM NaCl, 20 mM bis-tris, 10 mM HCl, 1 mM TCEP-HCl, 1 mM EDTA and 10% v/v glycerol. Final reaction conditions included 500 nM Usb1 and 100 nM RNA substrate in 50 mM NaCl, 50 mM KCl, 20 mM bis-tris, 10 mM HCl, 1 mM TCEP-HCl, 10% w/v sucrose, 5% v/v glycerol, 0.5 mM EDTA, 0.005% v/v Triton X-100 and 0.1 mg mL$^{-1}$ BSA. Samples were incubated at room temperature for the indicated time and reactions were quenched by the addition of an equal volume of 100% deionized formamide. Samples were resolved on a 20% 19:1 acrylamide:bis-acrylamide PAGE gel containing 8 M urea, 89 mM Tris borate and 2 mM EDTA. The gels were imaged directly through low fluorescence glass plates (CBS Scientific) using a Typhoon FLA 9000 (GE Healthcare Life Sciences). Alkaline hydrolysis ladders were produced by incubating 5 μL of RNA substrate (200 nM) in buffer with 5 μL of 50 mM bicarbonate buffer pH 9.2, 1 mM EDTA at 95 °C for 10 min.

Samples were treated with CIP or T4 PNK by addition of "Cutsmart" or "PNK" buffer from New England Biolabs and 10 units of CIP or T4 PNK and incubation at 37 °C for 15 min. Mock treated samples contained only Cutsmart buffer and water in lieu of CIP or T4 PNK.

For time-course experiments, the percent processed was calculated using the ratio of product(s) to total signal at 0, 5, 15, 30 and 60 min time points. Resulting data were fit to a one-phase exponential association equation $Y = Y_0 + (Y_{max} - Y_0) \times (1 - e^{-kx})$, where $Y$ is the % of the substrate processed, $x$ is time and $k$ is the rate constant (GraphPad Prism 4).

**Gel shift assay**. Gel shift assays were performed by incubating Cy3-labeled U6 with increasing concentrations of protein. Cy3-U6 RNA was heated to 90°C for 2 minutes in RNA binding buffer (100 mM KCl, 20% v/v sucrose, 20 mM bis-tris, 10 mM HCl, 1 mM EDTA, 1 mM TCEP-HCl, 0.01% Triton X-100, 0.2 mg mL$^{-1}$ tRNA, 0.02 mg mL$^{-1}$ sodium heparin), then snap cooled on wet ice. Proteins were prepared as 2x stocks in protein binding buffer (100 mM KCl, 20% sucrose v/v, 20 mM bis-tris, 10 mM HCl, 1 mM EDTA, 1 mM TCEP-HCl, 0.01% Triton X-100, 0.2 mg mL$^{-1}$ BSA). Final binding reactions contained 5 nM Cy3-U6, 100 mM KCl, 20% sucrose v/v, 20 mM bis-tris, 10 mM HCl, 1 mM EDTA, 1 mM TCEP-HCl, 0.01% Triton X-100, 0.1 mg mL$^{-1}$ tRNA, 0.01 mg mL$^{-1}$ sodium heparin, and 0.1 mg mL$^{-1}$ BSA. Samples were incubated at room temperature for 20 minutes prior to electrophoresis on a 6% polyacrylamide gel (29:1 acrylamide:bis-acrylamide, 89 mM Tris borate, 2 mM EDTA pH 8.0) for 2-3 hours at 3W at 4°C. Gels were imaged directly through low fluorescence glass plates (CBS Scientific) using a Typhoon FLA 9000 (GE Healthcare Life Sciences). Results were analyzed using ImageJ software and binding curves were fit using nonlinear regression in GraphPad Prism 4 to the Hill equation: % bound = $(B_{max}*[\text{protein}]^H)/(K_d^H + [\text{protein}]^H)$. Bmax was restrained to be between 0 and 100%, and Kd and H were restrained to be > 0.

**Fluorescence polarization binding experiments**. Fluorescence polarization binding reactions were performed by mixing 100 μL of 2× RNA in buffer (100 mM NaCl, 20 mM bis-tris, 10 mM HCl, 1 mM TCEP-HCl, 5% v/v glycerol and 1 mM EDTA) containing either 0.2 mg mL$^{-1}$ (for Lsm2–8 binding) or 0.02 mg mL$^{-1}$ (for Lhp1 binding) yeast tRNA (Roche Diagnostics) and 100 μL of protein in buffer containing 0.2 mg mL$^{-1}$ BSA (Ambion) in black 96-well microplates (Greiner Bio-One). Final RNA concentrations for Lsm2–8, Lhp1 and before/after Usb1 binding experiments were 0.25, 1 and 0.5 nM, respectively. Final protein concentrations were 0.001–1000 nM. For the before/after Usb1 binding experiment, FAM-U6 95–112+1U (100 nM) with a *cis*-diol was incubated with Usb1 1–290 (500 nM) as described above for 1 h before dilution and use in the binding experiments.

Fluorescence polarization was measured on a Tecan Infinite M1000Pro using an excitation wavelength of 470 nm and emission wavelength of 519 nm. Gain was optimized for each microplate. Fluorescence polarization was measured in triplicate for each condition (using 500–1000 flashes) and averaged. Data were normalized to the values for 0 nM protein (smallest value) and to the highest value, then averaged between three technical replicates. Binding curves were fit using nonlinear regression in GraphPad Prism 4 to a four-parameter logistic equation: % bound = $FP_{min} + (FP_{max} - FP_{min})/(1 + 10^{((\log K_d - \log[\text{protein}]) \times H)})$, where $FP_{min}$ and $FP_{max}$ are the normalized minimum and maximum % bound, $K_d$ is the binding dissociation constant and $H$ is the Hill coefficient. $FP_{min}$, $FP_{max}$, $K_d$ and $H$ were restrained to be >0.

**pH dependence exoribonuclease assay**. We found that Usb1 activity is strongly inhibited by the presence of either sulfate or phosphate. Thus, we sought to make a mixed buffer for pH-varied activity assays containing 200 mM each of sodium

acetate, bis-tris, sodium HEPES base, Tris base and CHES (pH~9). The pH of this 1 M solution was adjusted up or down with 5 M NaOH or 5 M HCl respectively. Aliquots were removed at every 0.5 pH unit step and kept as a 50× stock of buffer. Protein and RNA were diluted to 1 µM or 200 nM, respectively, in buffer containing 100 mM NaCl, 10% w/v sucrose, 0.1 mM EDTA, 0.1 mM TCEP, 0.1 mg mL$^{-1}$ BSA, 0.01% v/v Triton X-100 and 20 mM of the composite buffer. Protein and RNA were mixed together in equal volumes (for final concentrations of 500 and 100 nM respectively) for the indicated times. Reactions were quenched by addition of an equal volume of 100% formamide and samples were resolved as described above.

**Yeast strains and plasmids.** The *USB1* coding region and 500 bp up- and downstream of the coding region was amplified from genomic DNA from the BJ2168 (*MATa leu2 trp1 ura3-52 prb1-1122 pep4-3 prc1-407 gal2*) strain via PCR, which created an upstream BamHI site and a downstream *Xho* site. The PCR product was digested with BamHI and XhoI (New England Biolabs) and ligated into BamHI/XhoI-cut pRS416 and pRS414. Point mutations in pRS414-Usb1 were generated using inverse PCR as described above.

BJ2168 was transformed with pRS416-Usb1 using the lithium acetate method[63]. The Usb1 disruption strain (yTJC0700) was constructed by transformation of BJ2168 pRS416-Usb1 with linear DNA amplified from pAG32 (Euroscarf) containing the *hph* gene flanked by regions homologous to 500 nucleotides up- and downstream of the *USB1* gene. Cells were grown on YPD for 1 day, then replica plated onto media containing 200 µg mL$^{-1}$ hygromycin. Individual colonies were screened and deletion of *USB1* was confirmed via PCR.

Overexpression plasmids containing Usb1 were generated by PCR amplification of ORFs using primers that added an upstream BamHI and a downstream SalI site. The coding sequence for hUsb1 was codon optimized for expression in *S. cerevisiae* and the nucleotide sequence is listed in Supplementary Table 4. This PCR product was digested with BamHI and SalI (New England Biolabs) and ligated into BamHI/SalI-cut p425-GPD (ATCC 87359). All mutations, including addition of a single N-terminal hemagglutinin (HA) tag, were generated using inverse PCR as described above.

**USB1 complementation assays.** yTJC0700 was transformed with variants of pRS414-Usb1 or p425-GPD plasmids using the lithium acetate method[63]. Growth phenotypes were assessed by spotting 10-fold serial dilutions (OD$_{600}$=0.5, 0.05, 0.005) onto solid medium lacking tryptophan (for pRS414 plasmids) or leucine (for p425-GPD plasmids) and containing 1 mg mL$^{-1}$ 5-fluoroorotic (5FOA) acid. Plates were incubated at 30 °C for 3 days.

**Western blotting.** Yeast were grown in selective media and total protein was isolated by trichloroacetic acid precipitation[64]. Protein concentration was normalized by A$_{280}$ measurement and equivalent amounts were separated on a 4–20% Criterion TGX midi protein gel (200 V for 1 h; Bio-Rad) and subsequently transferred to a nitrocellulose membrane (30 min, 100 V, 4 °C). The membrane was blocked using 5% (w/v) nonfat dry milk and probed using an HA antibody conjugated to horseradish peroxidase used at 1:5,000 dilution (Sigma Aldrich 11667475001). Mouse α-actin (AMB Millipore MAB1501) was used at 1:5,000 dilution and goat α-mouse HRP antibodies were used at 1:10,000 dilution (Bio-Rad 1706515). Blots were developed using Clarity Western ECL substrate (Bio-Rad) and imaged using an ImageQuant LAS 4000 Imager (GE Healthcare Life Sciences).

**Data availability.** Coordinates and structure factors have been deposited in the PDB with accession codes 5UQJ and 5V1M. Other data supporting the findings of this manuscript are available from the corresponding author on reasonable request.

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

## Acknowledgements

We thank Kiyoshi Nagai, Yigong Shi and Stewart Shuman for plasmids and Jill Wildonger for the actin antibody. We thank Elsebet Lund and Jim Dahlberg for careful reading of the manuscript. We thank members of the Butcher, Brow, and Hoskins labs for helpful discussions. We thank Marco Tonelli and Mark Anderson for assisting with NMR data collection. This work was supported by grants from the US National Institutes of Health (R01 GM065166 to S.E.B. and D.A.B., R35 GM118131 to S.E.B. and R35 GM118075 to D.A.B., R00 GM086471 and R01 GM112735 to A.A.H., T32-GM08349 to T.J.C.). A.A.H. was supported by the Shaw Scientist Award, the Beckman Young Investigator Award and startup funding from the University of Wisconsin-Madison, Wisconsin Alumni Research Foundation (WARF), and the Department of Biochemistry. A.L.D. was supported by the University of Wisconsin-Madison Louis and Elsa Thomsen Wisconsin Distinguished Graduate Fellowship. T.J.C. was supported by the William H. Peterson Fellowship. A.T.D. was supported by the Hilldale Undergraduate Research Fellowship. S. L. was supported by a Sophomore Research Fellowship. This study made use of the National Magnetic Resonance Facility at Madison, which is supported by NIH grant P41GM103399 (NIGMS), old number: P41RR002301. Equipment was purchased with funds from the University of Wisconsin-Madison, the NIH P41GM103399, S10RR02781, S10RR08438, S10RR023438, S10RR025062, S10RR029220), the NSF (DMB-8415048, OIA-9977486, BIR-9214394) and the USDA. Use of the Advanced Photon Source, an Office of Science User Facility operated for the US Department of Energy (DOE) Office of Science by Argonne National Laboratory, was supported by the US DOE under contract no. DE-AC02-06CH11357. Use of the NE-CAT Sector 24 was supported by NIH grant P41 GM103403. Fluorescence polarization data were obtained at the University of Wisconsin-Madison Biophysics Instrumentation Facility, which was established with support from the University of Wisconsin-Madison and grants BIR-9512577 (NSF) and S10RR13790 (NIH).

## Author contributions

A.L.D., E.J.M., T.J.C. and S.E.B. designed research; A.L.D., E.J.M., T.J.C., A.T.D. and S.E. L. performed research. A.L.D. analyzed data. A.L.D., E.J.M. and A.T.D. prepared crystallization samples. A.L.D. and E.J.M. performed structure determination. A.L.D. and S.E.L. performed biochemical assays. A.L.D. and T.J.C. generated the Usb1 deletion yeast strain and carried out experiments. T.J.C. performed western blots. W.M.W. wrote NMR pulse programs and assisted with data collection and analysis. A.L.D. and S.E.B. wrote the paper with input from A.A.H., D.A.B., E.J.M. and T.J.C.

## Additional information

**Competing interests:** The authors declare no competing financial interests.

