## [Peer Review file · Nature Communications]

Reviewers' comments:

Reviewer #1 (Remarks to the Author):

The authors report the biochemical and structural characterization of the enzyme Usb1, which is involved in the processing of U6 snRNA, and they established an in vitro reconstitution system for the U6 snRNP from yeast resulting in a model for U6 snRNP assembly.

The authors determined the crystal structure of the yeast Usb1, which turned out to be very similar to the already known structure of human Usb1. In addition, they obtained a crystal structure of the human Usb1 with a bound 5'UMP.

All experiments regarding the enzymatic activities of yeast and human Usb1 are well done and reveal significant differences between the human and yeast enzymes. Also the binding and assembly assays are technically sound.

Minor points:

i) in order to solve the crystallographic phase problem the authors used a heavy atom derivative. Why did the authors not use Molecular Replacement using the structure of human Usab1? A comment on that in the Methods Section would be helpful.

ii) why did they authors use the human Usb1 to get a structure with a bound nucleotide? Does yeast Usb1 not bind the nucleotide? Or was it due to the fact that yeast Usb1 was crystallized in presence of 2M (NH₄)₂SO₄, while human Usb1 crystals could be transferred to 20% PEG? The authors should add one sentence to the results section and explain to the reader the rationale why using the human instead of the yeast protein.

iii) a major difference between yeast and human U6 snRNP assembly is caused by the different 3' ends of the U6 snRNA, which carries a 3'phosphate in yeast, while human U6 has a 2',3'-cyclic phosphate. Interestingly, the human and yeast Lsm2-8 proteins specifically recognize (or discriminate) these different 3'ends. The authors correctly state that the crystal structure of the yeast U6 snRNP can not easily explain this specificity, as the RNA is lacking a fifth nucleotide and the 3'phosphate.

But what about the cryo-EM structures of the U4/U6.U5 tri-snRNP? The two cryo-EM structures of yeast tri-snRNP (PDB 3JCM, 5GAN) are reported to have a resolution of 3.7 Å and 3.8 Å, respectively.

Reviewer #2 (Remarks to the Author):

This article characterizes the activity of yeast Usb1, a 3'-end RNA processing enzyme involved in U6 RNA maturation. It shows difference in activity with respect to the human enzyme, with an additional CPDase activity. A crystal structure of the human enzyme with a substrate analogue confirms that the binding sites of the human and yeast enzymes are very similar and supports the conclusion that residues outside the binding site determine the differences in the enzyme behaviors. However, no explanation is provided for the additional CPDase activity of the γ Usb1. Instead, the role of the aminoacid at position 84 and of the N-terminal part of the enzyme are analysed. Further, the article investigates the RNA binding capability of Lsm2-8 and Lhp1, two additional U6 maturation factors. A model is proposed for the sequence of maturation events in U6 RNA processing.

The article has some nice piece of data and some interesting results (such as the specificity of Lsm2-8 and Lhp1 for certain RNAs and the different activities of Usb1). However, it is difficult to follow a punch line from beginning to end; rather than a full story it reads as a collection of

important and useful findings. The article contains relevant information but is not easy to read through, as it misses a clear underlining story that would keep up attention.

A couple of specific points:

The gain of function in the hUsb1-H84F enzyme in yeast is explained by a better stacking of the phenylalanine with the n-1 nucleotide. Why is this not relevant in humans? The available crystal structure, from which the hypothesis is derived, is for the hUsb1.

What is the reason for the different pH dependence of the human and yeast enzyme? On the basis of the crystal structures it should be possible to formulate a hypothesis.

Any ideas what are the structural determinants of the additional CPDase activity in yeast?

Does yeast Lsm2-8 bind to a 2'-3' cyclic phosphate and if so with which affinity? Does human Lsm2-8 bind to a 3' phosphate and if so with which affinity?

In the discussion it is stated "Humans also possess a counteracting TUTase that extends the 3' tail of U6. *S. cerevisiae* has no identifiable TUTase and thus the autoinhibition of γ Usb1 by its own product is necessary to prevent over-processing of U6." It is not clear to me whether the authors imply that the CPDase activity of the enzyme improves its autoinhibition activity, as opposed to hUsb1. If so, how?????

Reviewer #1 (Remarks to the Author):

The authors report the biochemical and structural characterization of the enzyme Usb1, which is involved in the processing of U6 snRNA, and they established an in vitro reconstitution system for the U6 snRNP from yeast resulting in a model for U6 snRNP assembly.

The authors determined the crystal structure of the yeast Usb1, which turned out to be very similar to the already known structure of human Usb1. In addition, they obtained a crystal structure of the human Usb1 with a bound 5'UMP.

All experiments regarding the enzymatic activities of yeast and human Usb1 are well done and reveal significant differences between the human and yeast enzymes. Also the binding and assembly assays are technically sound.

We thank the reviewer for the positive comments.

Minor points:

i) in order to solve the crystallographic phase problem the authors used a heavy atom derivative. Why did the authors not use Molecular Replacement using the structure of human Usb1? A comment on that in the Methods Section would be helpful.

Initial attempts using molecular replacement did not give useable phases for subsequent map calculation and model building, due to the low sequence identity (18%) between the two proteins. Therefore, a heavy atom derivative was used in order to determine initial phases for map calculation and model building. A comment in the Methods section (pg. 12) has been added.

ii) Why did they authors use the human Usb1 to get a structure with a bound nucleotide? Does yeast Usb1 not bind the nucleotide? Or was it due to the fact that yeast Usb1 was crystallized in presence of 2M (NH₄)₂SO₄, while human Usb1 crystals could be transferred to 20% PEG? The authors should add one sentence to the results section and explain to the reader the rationale why using the human instead of the yeast protein.

The reviewer is correct in that attempts to soak in nucleotides to ScUsb1 crystals were unsuccessful, likely due to the stringent requirement for 2M ammonium sulfate in the crystallization conditions. A statement about this in the Methods section has been added on pg. 13.

iii) A major difference between yeast and human U6 snRNP assembly is caused by the different 3' ends of the U6 snRNA, which carries a 3' phosphate in yeast, while human U6 has a 2',3'-cyclic phosphate. Interestingly, the human and yeast Lsm2-8 proteins specifically recognize (or discriminate) these different 3'ends. The authors correctly state that the crystal structure of the yeast U6 snRNP cannot easily explain this specificity, as the RNA is lacking a fifth nucleotide and the 3' phosphate. But what about the cryo-EM structures of the U4/U6.U5 tri-snRNP? The two cryo-EM structures of yeast tri-snRNP (PDB 3JCM, 5GAN) are reported to have a resolution of 3.7 Å and 3.8 Å, respectively.

We stated on pg. 2 that "the position of the terminal phosphate on yeast U6 RNA (at either the 2' or 3' oxygen) was not known, and cannot be resolved in recent cryo-EM structures¹⁹⁻²⁴". The resolution in recent spliceosome structures are reported for the best-

resolving cores of the structure. In the tri-snRNP structure (PDB 5GAN), the 3' tail of U6/Lsm2-8 is the worst-resolving part of the structure, with a resolution of 7.5-10 Å (see Nguyen *et al.* Nature 2016 extended data figure 2). The 3.8 Å structure from the Shi group (PDB 3JCM) suffers from the same problem. Structures of active spliceosome complexes (B^{act}/C/C*/ILS) lack Lsm2-8 and suffer from a poorly-resolved U6 3' tail. We have added a comment on pg. 2 to address this.

Reviewer #2 (Remarks to the Author):

This article characterizes the activity of yeast Usb1, a 3'-end RNA processing enzyme involved in U6 RNA maturation. It shows difference in activity with respect to the human enzyme, with an additional CPDase activity. A crystal structure of the human enzyme with a substrate analogue confirms that the binding sites of the human and yeast enzymes are very similar and supports the conclusion that residues outside the binding site determine the differences in the enzyme behaviors. However, no explanation is provided for the additional CPDase activity of the yUsb1. Instead, the role of the amino acid at position 84 and of the N-terminal part of the enzyme are analyzed. Further, the article investigates the RNA binding capability of Lsm2-8 and Lhp1, two additional U6 maturation factors. A model is proposed for the sequence of maturation events in U6 RNA processing. The article has some nice piece of data and some interesting results (such as the specificity of Lsm2-8 and Lhp1 for certain RNAs and the different activities of Usb1). However, it is difficult to follow a punch line from beginning to end; rather than a full story it reads as a collection of important and useful findings. The article contains relevant information but is not easy to read through, as it misses a clear underlining story that would keep up attention.

We thank the reviewer for the positive comments and hope these revisions help clarify the underlying story of how the enzymatic activity of Usb1 controls downstream U6 snRNP assembly.

A couple of specific points:

The gain of function in the hUsb1-H84F enzyme in yeast is explained by a better stacking of the phenylalanine with the n-1 nucleotide. Why is this not relevant in humans? The available crystal structure, from which the hypothesis is derived, is for the hUsb1.

In our human Usb1-5'UMP structure, we have the “n” nucleotide bound in the active site (as 5' UMP) but cannot tell where the n-1 nucleotide stacks. We agree with the reviewer that stacking should also be possible with a histidine at position 84 and have therefore deleted the confusing hypothesis that H84F is consistent with stacking of the n-1 nucleobase (pg. 6). Although we do not yet understand the physical basis for the hUsb1-H84F gain of function mutation, we feel the result is important because it allows the human enzyme to complement the yeast deletion strain and enables us to show that the N-terminal region of the enzyme has a conserved essential function *in vivo* (Figure 4D).

What is the reason for the different pH dependence of the human and yeast enzyme? On the basis of the crystal structures it should be possible to formulate a hypothesis.

As discussed on pg. 5, the difference in pH dependence for human and yeast enzymes is likely due to differences in pK_a of the active site histidines. As requested, we have added the following hypothesis on pg. 5: “This difference in pH optimum suggests that the active site histidine residues in yeast and hUsb1 have markedly different pK_a 's, likely due to different active

site microenvironments. For example, human Usb1 has a histidine (H84) adjacent to the active site that can hydrogen bond to the active site serine S122.¹¹ In yeast, this position is a phenylalanine (F78) which cannot form an analogous hydrogen bond.”

Any ideas what are the structural determinants of the additional CPDase activity in yeast?

We agree with the reviewer that pinpointing the structural determinants of CPDase activity is very interesting. Such an endeavor would likely require high resolution structures of complexes bound to a transition state analogue and/or cyclic phosphate product, along with computational simulations. We note that RNase A is arguably the most well studied enzyme (as detailed in hundreds of papers over the past 60 years), and yet has a mechanism that, until recently, was still being debated (see for example Elsässer, Fels and Weare, JACS 2013). Our investigations into the origin of yeast Usb1 CPDase activity are ongoing.

Does yeast Lsm2-8 bind to a 2'-3' cyclic phosphate and if so with which affinity? Does human Lsm2-8 bind to a 3' phosphate and if so with which affinity?

To address this question, we have included new data measuring yeast Lsm2-8 and Lhp1 binding to full-length RNAs containing a cyclic phosphate and compare this to the cis-diol (Supplementary Figure S6). These data reveal that yeast Lsm2-8 binds significantly more tightly to the cis diol than the cyclic phosphate. In contrast, the Bindereif group has tested human Lsm2-8 binding to a 2',3'-cyclic phosphate and cis diol and showed the opposite, that the cyclic phosphate improves affinity for human Lsm2-8 (Licht *et al.* RNA 2008). Therefore, the Lsm2-8 complex has clearly evolved to bind the Usb1-processed U6 RNA product. To our knowledge no one has investigated human Lsm2-8 binding to a 3' phosphate containing RNA, presumably because this modification does not occur in humans. This might be an interesting future experiment although we feel it is outside the scope of our current study as we are not investigating the binding properties of the human Lsm2-8 complex.

In the discussion it is stated “Humans also possess a counteracting TUTase that extends the 3' tail of U6. *S. cerevisiae* has no identifiable TUTase and thus the autoinhibition of yUsb1 by its own product is necessary to prevent over-processing of U6.” It is not clear to me whether the authors imply that the CPDase activity of the enzyme improves its autoinhibition activity, as opposed to hUsb1. If so, how?????

We state on pg. 4 and graphically in Figure 1E that “These data demonstrate that yUsb1 is incapable of further processing its dominant 3' phosphate product... The presence of both exonuclease and CPDase activity, along with inactivity on 3' phosphate-terminated substrates, reveals an elegant mechanism for ensuring that yUsb1 does not over process and degrade U6 RNA.” Thus we have directly demonstrated that CPDase activity in yUsb1 prevents overprocessing via autoinhibition. The mechanism by which CPDase activity inhibits further processing is an interesting question. We have provided a sentence in the discussion (pg. 10) speculating that “A 3' phosphate might encounter steric clash with nearby loop residues of Usb1.”

REVIEWERS' COMMENTS:

Reviewer #1 (Remarks to the Author):

The authors have addressed all questions in a very satisfying way. Hence, I recommend publication of the manuscript as it is.

Reviewer #2 (Remarks to the Author):

The authors have answered all my specific comments satisfactorily.

I still find the paper difficult to read: the amount of data is overwhelming and probably could be presented in a more attractive way. However, this is something that the journal editor should decide upon. From the scientific/technical point of view, the work is sound and I have no further questions.